# Learning compositional sequences with multiple time scales through a hierarchical network of spiking neurons

**Amadeus Maes**[1], **Mauricio Barahona**[2], **Claudia Clopath**[1]*

**1** Bioengineering Department, Imperial College London, London, United Kingdom, **2** Mathematics Department, Imperial College London, London, United Kingdom

* c.clopath@imperial.ac.uk

**Data Availability Statement:** The code is available online at url: https://github.com/amaes-neuro/compositional-sequences.

## Abstract

Sequential behaviour is often compositional and organised across multiple time scales: a set of individual elements developing on short time scales (motifs) are combined to form longer functional sequences (syntax). Such organisation leads to a natural hierachy that can be used advantageously for learning, since the motifs and the syntax can be acquired independently. Despite mounting experimental evidence for hierarchical structures in neuroscience, models for temporal learning based on neuronal networks have mostly focused on serial methods. Here, we introduce a network model of spiking neurons with a hierarchical organisation aimed at sequence learning on multiple time scales. Using biophysically motivated neuron dynamics and local plasticity rules, the model can learn motifs and syntax independently. Furthermore, the model can relearn sequences efficiently and store multiple sequences. Compared to serial learning, the hierarchical model displays faster learning, more flexible relearning, increased capacity, and higher robustness to perturbations. The hierarchical model redistributes the variability: it achieves high motif fidelity at the cost of higher variability in the between-motif timings.

## Author summary

The brain has the ability to learn and execute sequential behaviour on multiple time scales. This behaviour is often compositional: a set of simple behaviours is concatenated to create a complex behaviour. Technological improvements increasingly shine light on the building blocks of compositional behaviour, yet the underlying neural mechanisms remain unclear. Here, we propose a hierarchical model to study the learning and execution of compositional sequences, using bio-plausible neurons and learning rules. We compare the hierarchical model with a serial version of the model. We demonstrate that the hierarchical model is more flexible, efficient and robust by exploiting the compositional nature of the sequences.

**Funding:** AM acknowledges funding through the EPSRC Centre for Neurotechnology (https://epsrc.ukri.org/skills/students/centres/2013-cdt-exercise/neurotechnologyforlifeandhealth/). MB acknowledges funding through EPSRC award EP/N014529/1 supporting the EPSRC Centre for Mathematics of Precision Healthcare at Imperial (https://www.imperial.ac.uk/mathematics-precision-healthcare). CC acknowledges support by BBSRC BB/N013956/1 (https://bbsrc.ukri.org/), BB/N019008/1, Wellcome Trust 200790/Z/16/Z (https://wellcome.org/?gclid=CjwKCAjw5Kv7BRBSEiwAXGDElTTydWe4MWJu_2waXdH7DsTdOym3ijPyGHfHePBEuai0XKfJa5RbIRoC3KcQAvD_BwE), Simons Foundation 564408 (https://www.simonsfoundation.org/) and EPSRC EP/R035806/1 (https://epsrc.ukri.org/). The funders had no role in study design, data collection and analysis, decision to publish, or preparation of the manuscript.

**Competing interests:** The authors have declared that no competing interests exist.

## Introduction

Many natural behaviours are compositional: complex patterns are built out of combinations of a discrete set of simple motifs [1–3]. Compositional sequences unfolding over time naturally lead to the presence of multiple time scales—a short time scale is associated with the motifs and a longer time scale is related to the ordering of the motifs into a syntax. How such behaviours are learnt and controlled is the focus of much current research. Broadly, there are two main strategies for the modeling of sequential behaviour: serial and hierarchical. In a serial model, the long-term behaviour is viewed as a chain of motifs proceeding sequentially, so that the first behaviour in the chain leads to the second and so on ('domino effect'). Serial models present some limitations [4, 5]. Firstly, serial models have limited flexibility since relearning the syntax involves rewiring the chain. Secondly, such models lack robustness, e.g., breaking the serial chain halfway means that the later half of the behaviour is not produced. It has been proposed theoretically that hierarchical models can alleviate these problems, at the cost of extra hardware.

Evidence for the presence of hierarchical structures in the brain is mounting [6–8]. Furthermore, experiments are increasingly shining light on the hierarchical mechanisms of sequential behaviour. An example is movement sequences in multiple animal models, such as *Drosophila* [9–11], mice [12–14] and *C. elegans* [15, 16]. Simultaneous recordings of behaviour and neural activity are now possible in order to relate the two together [17, 18]. Songbirds are another example of animals that produce stereotypical sequential behaviour: short motifs are strung together to form songs. In this case, a clock-like dynamics is generated in the premotor nucleus HVC of the bird's brain, such that neurons are active in sequential bursts of $\sim 10$ ms [19]. This activity is thought to control the timing of the spectral content of the song (the within-motif dynamics). The between-motif dynamics has a different temporal structure [20, 21]; hence the ordering of the motifs into a song (the syntax) might be controlled by a different mechanism. Supporting this view, it has been found that learning the motifs and syntax involves independent mechanisms [22]. The computational study of hierarchical structures and compositional behaviour can also lead to insights into the development of human locomotion and language as there are striking conceptual parallels [23–26].

Here, we present a model for learning temporal sequences on multiple scales implemented through a hierarchical network of bio-realistic spiking neurons and synapses. In contrast to current models, which focus on acquiring the motifs and speculate on the mechanisms to learn a syntax [27–29], our spiking network model learns motifs and syntax independently from a target sequence presented repeatedly. Furthermore, the plasticity of the synapses is entirely local, and does not rely on a global optimisation such as FORCE-training [30–32] or backpropagation through time [33]. To characterise the effect of the hierarchical organisation, we compare the proposed hierarchical model to a serial version by looking at their learning and relearning behaviours. We show that, contrary to the serial model, the hierarchical model acquires the motifs independently from the syntax. In addition, the hierarchical model has a higher capacity and is more resistant to perturbations, as compared to a serial model. We also investigate the variability of the neural activity in both models, during spontaneous replay of stored sequences. The organisation of the model shapes the neural variability differently. The within-motif spiking dynamics is less variable in a hierarchical organisation, while the time between the execution of motifs is more variable.

The paper is organised as follows. We start by describing the proposed hierarchical spiking network model and the learning protocol. We then analyse the learning and relearning behaviour of the proposed model, and compare it to the corresponding serial model. Next, we investigate several properties of the model: (i) the performance and consistency of spontaneous

sequence replays on a range of learnt sequences; (ii) capacity, i.e., how multiple sequences can be stored simultaneously; (iii) robustness of the sequence replays.

## Results

### Hierarchical model of spiking neurons with plastic synapses for temporal sequence learning

We design a hierarchical model by combining the following spiking recurrent networks (Fig 1): 1) A recurrent network exhibiting fast sequential dynamics (the *fast clock*); 2) a recurrent network exhibiting slow sequential dynamics (the *slow clock*); 3) a series of interneuron networks that store and produce the to-be-learnt ordering of motifs (the *syntax networks*); 4) a series of read-out networks that store and produce the to-be-learnt motif dynamics (the *motif networks*). We assume that there are a finite number of motifs and each motif is associated to a separate read-out network (e.g., in Fig 1 there are 2 read-out networks corresponding to motifs *A* and *B*). The goal of the model is to learn a complex sequence, with the motifs arranged in a certain temporal order, such that the motifs themselves and the temporal ordering of the motifs are learnt using local plasticity rules.

**Neuronal network architecture.** All neurons are either excitatory or inhibitory. Excitatory neurons follow an adaptive exponential integrate-and-fire dynamics and inhibitory neurons follow a standard integrate-and-fire dynamics (see Methods).

The model has two recurrent networks that exhibit sequential dynamics: the fast and slow clocks. The design of the clock networks follows Ref. [29]. Each clock is composed of clusters of excitatory neurons coupled in a cycle with a directional bias (i.e., neurons in cluster *i* are more strongly connected to neurons in cluster *i* + 1) together with a central cluster of

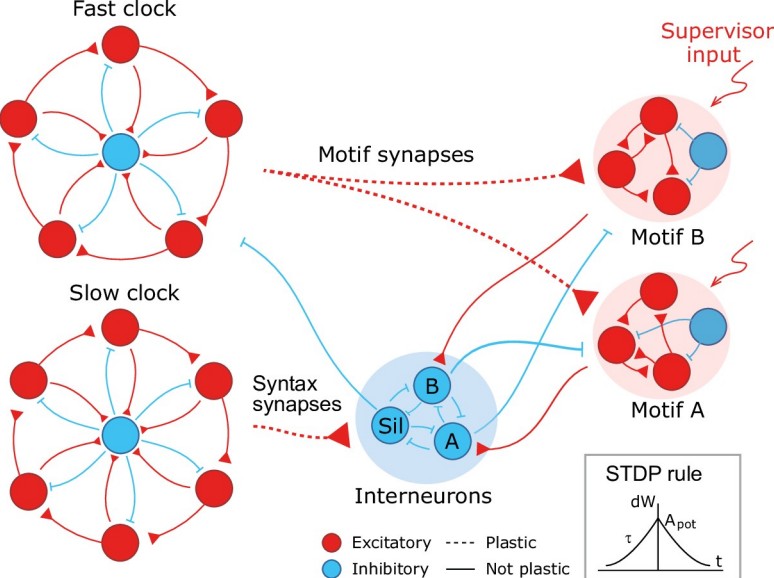

**Fig 1. A cartoon of the model.** Dynamics in the read-out networks (A and B) is learnt and controlled on two time scales. The fast time scale network (fast clock) exhibits sequential dynamics that spans individual motifs. This acts directly on the read-out networks through plastic synapses. These synapses learn the *motifs*. The slow time scale network (slow clock) exhibits sequential dynamics that spans the entire sequence of motifs. This acts indirectly on the read-out networks through an interneuron network. The synapses from the slow clock to the interneurons are plastic and learn the right order of the motifs, or the *syntax*. The plastic synapses follow a simple symmetric STDP rule for potentiation, with a constant depression independent of spike time.

inhibitory neurons coupled to all the excitatory clusters (Fig 1). This architecture leads to sequential dynamics propagating around the cycle and the period can be tuned by choosing different coupling weights. The individual motifs are not longer than the period of the fast clock and the total length of the sequence is limited to the period of the slow clock. In our case, we set the coupling weights of the fast clock such that a period of $\sim$ 200 ms is obtained, whereas the weights of the slow clock are set to obtain a period of $\sim$ 1000 ms.

The fast clock neurons project directly onto the read-out networks associated with each motif, which are learnt and encoded using a supervisor input. Hence the fast clock controls the within-motif dynamics. The slow clock neurons, on the other hand, project onto the interneuron network of inhibitory neurons. The interneuron network is also composed of clusters: there is a cluster associated with each motif, with coupling weights that inhibit all other motif networks, and one cluster associated with the 'silent' motif, with couplings that inhibit all motif networks and the fast clock. Hence the temporal ordering of the motifs (the syntax) can be encoded in the mapping that controls the activity of the interneurons driven by the slow clock. As a result of this hierarchical architecture, the model allows for a dissociation of within-motif dynamics and motif ordering. The two pathways, from the fast clock to the read-out and from the slow clock to the interneurons, each control a different time scale of the spiking network dynamics.

**Plasticity.**   Learning is accomplished through plastic synapses under a simple biophysically plausible local STDP rule (see Methods) governing the synapses from the fast clock to the read-out networks (motif synapses) and from the slow clock to the interneurons (syntax synapses). The STDP rule has a symmetric learning window and implements a Hebbian 'fire together, wire together' mechanism.

All other weights in the model are not plastic and are fixed prior to the learning protocol. The weights in the fast and slow clocks and the interneuron wiring are assumed to originate from earlier processes during evolution or early development. Previous computational studies have shown that sequential dynamics can be learnt in recurrent networks, both in an unsupervised [34, 35] and supervised [29, 36] fashion.

**Learning scheme.**   During learning, a target sequence is presented. We design a target sequence by combining motifs in any order, e.g., *AAB*. A time-varying external current, corresponding to the target sequence, projects to the excitatory neurons in the read-out networks. Additionally, a short external current activates the first cluster in the fast clock to signal the onset of a new motif (see Methods for more details). During the presentation of the target sequence, the plastic synapses change. When no target sequence is presented, spontaneous dynamics is simulated. Spontaneous dynamics replays the stored sequence. In this case, there is only random external input and no external input corresponding to a target sequence.

## The model allows for independent learning of motifs and syntax

We first show how a non-trivial sequence can be learned emphasising the role that each network plays. As an example, consider the target sequence *AAB*. This sequence is non-trivial as both the within-motif dynamics and syntax is non-Markovian (Fig 2A). Non-Markovian sequences are generally hard to learn, because they require a memory about past dynamics [37]. The sequential dynamics in the fast and slow clock provide a mechanism to overcome this challenge: by providing time indices the non-Markovian sequence is essentially transformed into a Markovian sequence. First, we present the target sequence repeatedly to the read-out networks (as shown in S1 Fig). After learning is finished, we test whether learning was successful by checking that the sequence is correctly produced by spontaneous dynamics (Fig 2B–2E). Note that the slow clock spans the entire sequence (Fig 2C) and activates the

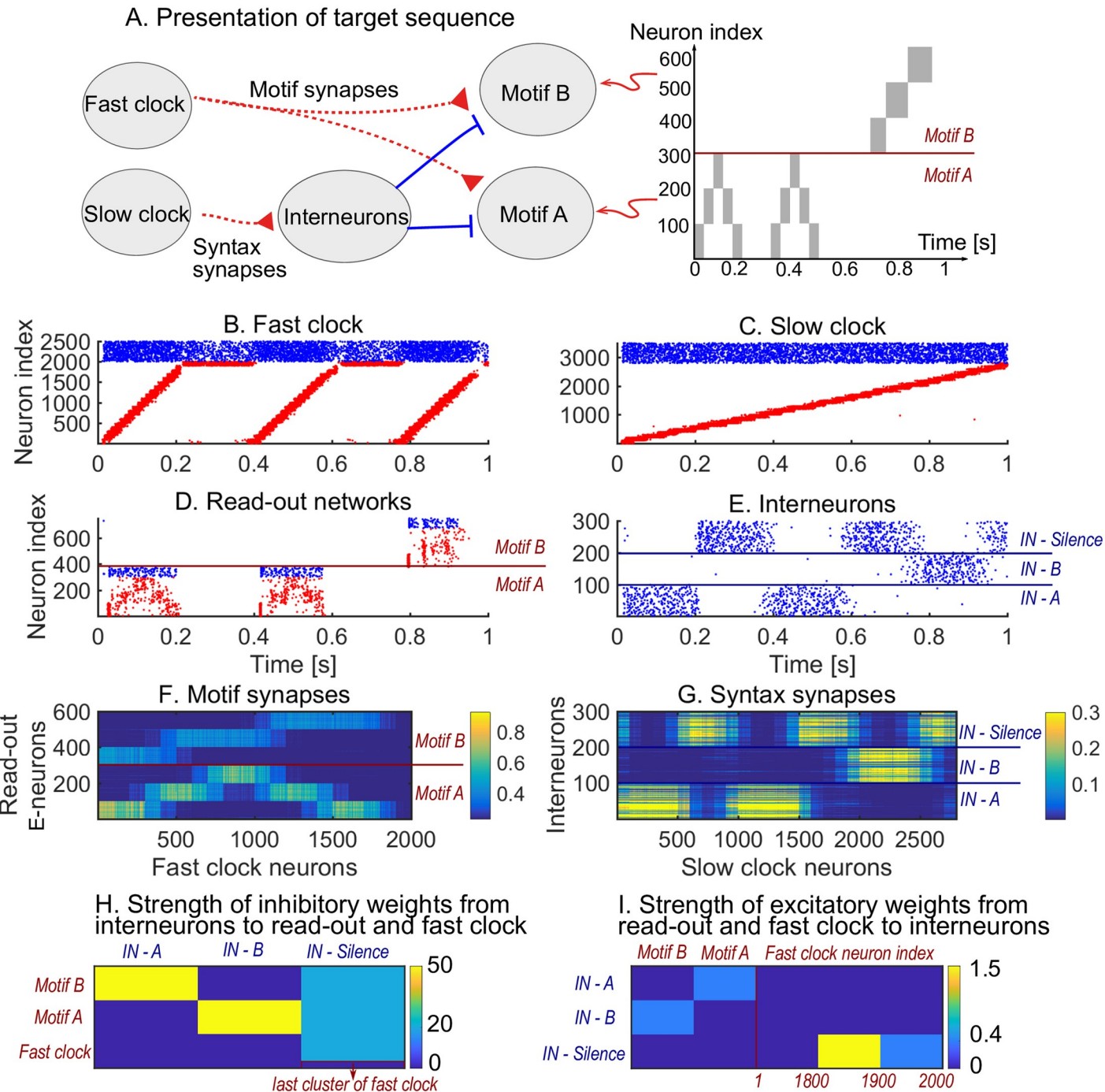

**Fig 2. Learning sequence *AAB*.** A. The target sequence is repeatedly presented to the read-out networks corresponding to motifs *A* and *B*. *A* and *B* are 200 ms long motifs. Between the motifs, we assume a silent period of 150 ms. B-E. Spontaneous dynamics after learning (50 target presentations). Red dots: excitatory neurons; blue dots: inhibitory neurons. B. The fast clock, controlled by interneurons 201 to 300. C. The slow clock, spanning and driving the entire sequence replay. D. The read-out networks, driven by the fast clock and controlled by the interneurons. E. The interneurons, driven by the slow clock. Neurons $1 - 100$ inhibit motif *B*. Neurons $101 - 200$ inhibit motif *A*. Neurons $201 - 300$ shut down both the fast clock and read-out networks. F. The motif synapses show that the target motifs *A* (neurons $1 - 300$ on the y-axis) and *B* (neurons $301 - 600$ on the y-axis) are stored. The weights for motif *A* are stronger because there are two *A*s in the target sequence and only one *B*. G. The syntax weights store the temporal ordering *A*-silent-*A*-silent-*B*-silent. H. Non-plastic inhibitory weights from the interneuron network to the read-out network and fast clock. I. Non-plastic excitatory weights from the read-out network and fast clock to the interneuron network.

interneurons in the correct order ([Fig 2E]), whereas the interneuron dynamics in turn determines the activation of the fast clock ([Fig 2B]) and the selection of a read-out network ([Fig 2D]). Through the learning phase, the motif weights (from the fast clock to the read-out networks) evolve to match the target motifs ([Fig 2F]), and, similarly, the syntax weights (from the slow clock to interneurons) evolve to match the ordering of the motifs in the target sequence ([Fig 2G]). Crucially, as shown below, these two sets of plastic weights are dissociated into separate pathways so that compositional sequences can be learnt efficiently through this model. The interneuron network coordinates the two pathways using non-plastic lateral inhibition ([Fig 2H]) and receives non-plastic excitatory input from the fast clock and motif networks ([Fig 2I]). Note that this conceptual model can be implemented in various ways (see [Methods] section Changing the slow clock into an all-inhibitory network) but can serve as a general framework for the learning and replay of stereotypical compositional behaviour.

### The hierarchical model enables efficient relearning of the syntax

We next demonstrate the ability of the model to relearn the ordering of the motifs. In general, we wish relearning to be efficient, i.e., the model should relearn the syntax without changing the motifs themselves. To test this idea, we perform a re-learning scheme $AAB \rightarrow ABA$ ([Fig 3]). An efficient model would only learn the switch in the syntax without the need to relearn the two motifs $A$ and $B$. Starting from a network where no sequence was stored, we begin with a learning phase where the sequence $AAB$ is presented (as in [Fig 2]) until it is learnt. We then switch to presenting the sequence $ABA$ in the relearning phase. To quantify the progress of learning throughout both phases, we simulate spontaneous dynamics after every fifth target sequence presentation and compute the error between the spontaneous dynamics and the target sequence (see [Methods] and [Fig 3]).

Our results show that the motifs are not re-learnt when switching between the first and second target sequences—the within-motif error keeps decreasing after we switch to the relearning phase indicating that there continues to be improved learning of the motifs common to both target sequences ([Fig 3A]). In contrast, the model relearns the temporal ordering of the

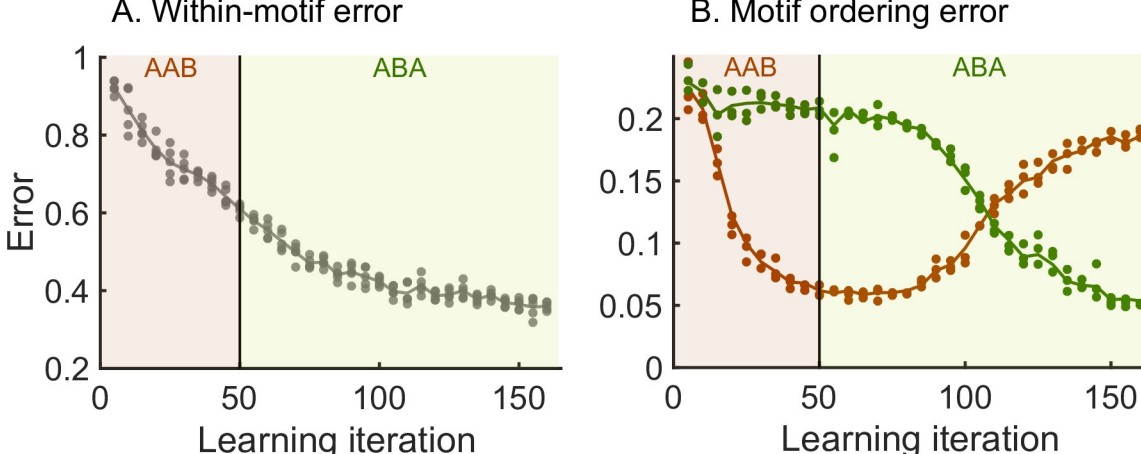

**Fig 3. Relearning syntax: *AAB → ABA*.** Brown shaded areas: presentation of target sequence *AAB*; dark green shaded areas: presentation of target sequence *ABA*. Brown dots: spontaneous dynamics is simulated 3 times, and the error with respect to the target sequence *AAB* is measured; dark green dots: spontaneous dynamics is simulated 3 times, and the error with respect to the target sequence *ABA* is measured. Lines guide the eye and are averages of the dots. See the [Methods] section for the details of the error measurements. A. The within-motif error keeps decreasing independent of the motif ordering. B. The motif ordering error (syntax error) switches with a delay.

motifs after the switch to the new target sequence—the syntax error relative to *AAB* decreases during the learning phase and then grows during relearning at the same time as the syntax error relative to *ABA* decreases (Fig 3B). Therefore, the hierarchy of the model allows for efficient relearning: previously acquired motifs can be reordered into new sequences without relearning the motifs themselves.

To investigate the role of the hierarchical organisation, we next studied how the relearning behaviour compares to a *serial model* with no dissociation between motifs and syntax. The serial model contains only one clock network and the read-out networks associated with each motif, with no interneurons (S2 Fig). In this serial architecture, motifs and syntax are both learnt and replayed by a single pathway (S2 Fig), and, consequently, when relearning the syntax, the motifs are also re-learnt from scratch even when there is no change within the individual motifs. This leads to a slower decay of the sequence error during learning and relearning in the serial model as compared to the hierarchical model (S3 Fig). The speed by which an old syntax is unlearned, and a new syntax is learned, depends on the learning rate of the syntax plasticity (S4 Fig).

The above results illustrate the increased efficiency of the hierarchical model to learn compositional sequences. The separation of motifs and syntax into two pathways, each of them associated with a different time scale and reflected in the underlying neuronal architecture, allows for the learning and control of the different aspects of the sequence independently.

## The hierarchical organisation leads to improved learning speed and high motif fidelity

We now study the effects of having a hierarchical organisation on the speed and quality of learning. To do so, we consider three target sequences of increasing complexity, where each target sequence is comprised of a motif presented three times (Fig 4A).

First, we studied the speed at which the pattern is learnt by the hierarchical model as compared to the serial model. The hierarchical model is roughly three times faster than the serial model in learning patterns consisting of three repetitions (Fig 4B). This is expected: in the hierarchical model, the same motif synapses are potentiated three times during a single target presentation, whereas no such repeated learning takes place in the serial model. Furthermore, the speed and quality of the learning also depends on the complexity of the target sequence, i.e., target sequences with rapid temporal changes are harder to learn. Learning target sequences with faster-changing, more complex temporal features leads to a degradation of the performance of both models, but the hierarchical model consistently learns roughly three times faster than the serial model for all patterns (Fig 4, left to right).

Another important quality measure of learning is the reliability and consistency of the pattern replayed by the model under spontaneous dynamics. To study this, we generated repeated trials in which the three target sequences learnt (in Fig 4) were replayed spontaneously, and we compared the variability of the read-out dynamics across the trials for both the hierarchical and serial models. We first computed the within-motif and between-motif variability in the spontaneous trials. The hierarchical model leads to low within-motif variability and higher variability in the between-motif timings. This follows from the spiking in the read-out networks, with highly variable time gaps between motifs in the hierarchical model (Fig 5A). On the other hand, the spike trains within the three motifs correlate strongly with each other for the hierarchical model (Fig 5B). This is the case for the three target sequences.

We then studied the consistency of the motif as it is repeated (three times) within a target sequence. We observe that a high degradation of the repeated motif towards the end of the sequence in the serial model, which is milder in the hierarchical model (Fig 5C). In summary,

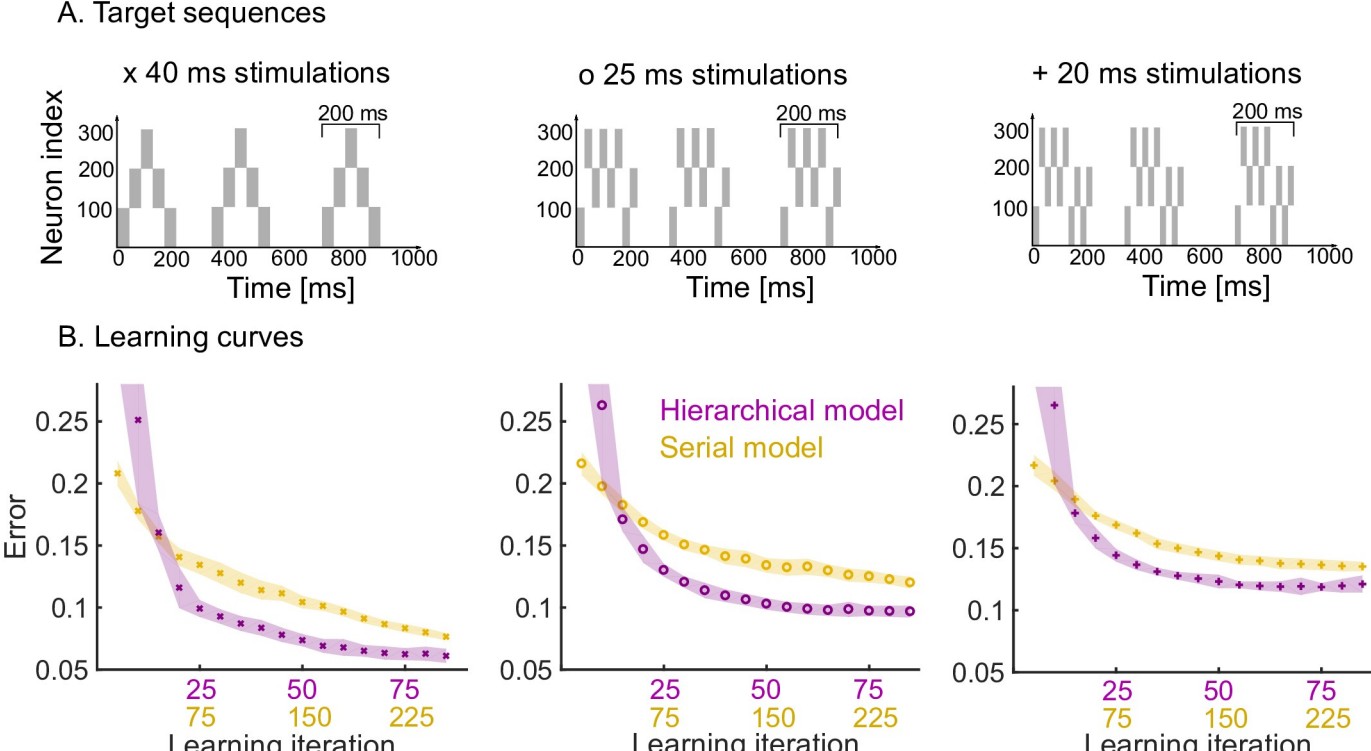

**Fig 4. Learning speed and performance of hierarchical and serial models on three target sequences of increasing temporal complexity.** A. Each target sequence consists of three presentations of the same motif (200 ms long) but with increasing complexity from left to right. Left: the simplest motif consists of five 40 ms stimulations. Middle: the motif consists of eight 25 ms stimulations. Right: the motif consists of ten 20 ms stimulations. B. Learning curves for the three target sequences for both the hierarchical and serial models. The same plasticity parameters are used for both models (see Methods). The shaded area indicates one standard deviation from the mean (50 trials). Note that the x-axis has two scales to show the three-fold increase in learning speed of the hierarchical model (i.e., for each learning iteration of the hierarchical model there are three iterations of the serial model). The performance degrades from left to right, as a more difficult target sequence is presented.

the hierarchical model produces accurate motifs that persist strongly over time, but with higher variability in the timing between them. The high reliability of the motifs is due to the stronger learning on the motif synapses discussed above. The higher variability in the inter-motif times is a result of the underlying variability of the periods of the clock networks. As discussed in Ref. [29], the sequential dynamics that underpins the clock networks operates by creating clusters of neurons that are active over successive periods of time. In that sense, the network uses neuron clusters to discretise a span of time (its period) into time increments. The variability of the period of the clock depends on the number of clusters, the number of neurons per cluster in the network, and the time span to discretise. A fast clock will thus have low variability in its period, whereas the slow clock is highly variable. The variability of the period of the serial clock is between the fast and slow clocks (Fig 5D). Consequently, within-motif temporal accuracy is maintained quasi-uniformly over the sequence in a hierarchical model. The price to pay is the addition of appropriately wired interneurons.

## The hierarchical organisation reduces the resources needed to store multiple sequences

As shown above, the plasticity of the model allows it to relearn single sequences, yet the relearning process might be too slow for particular situations. In general, animals acquire and

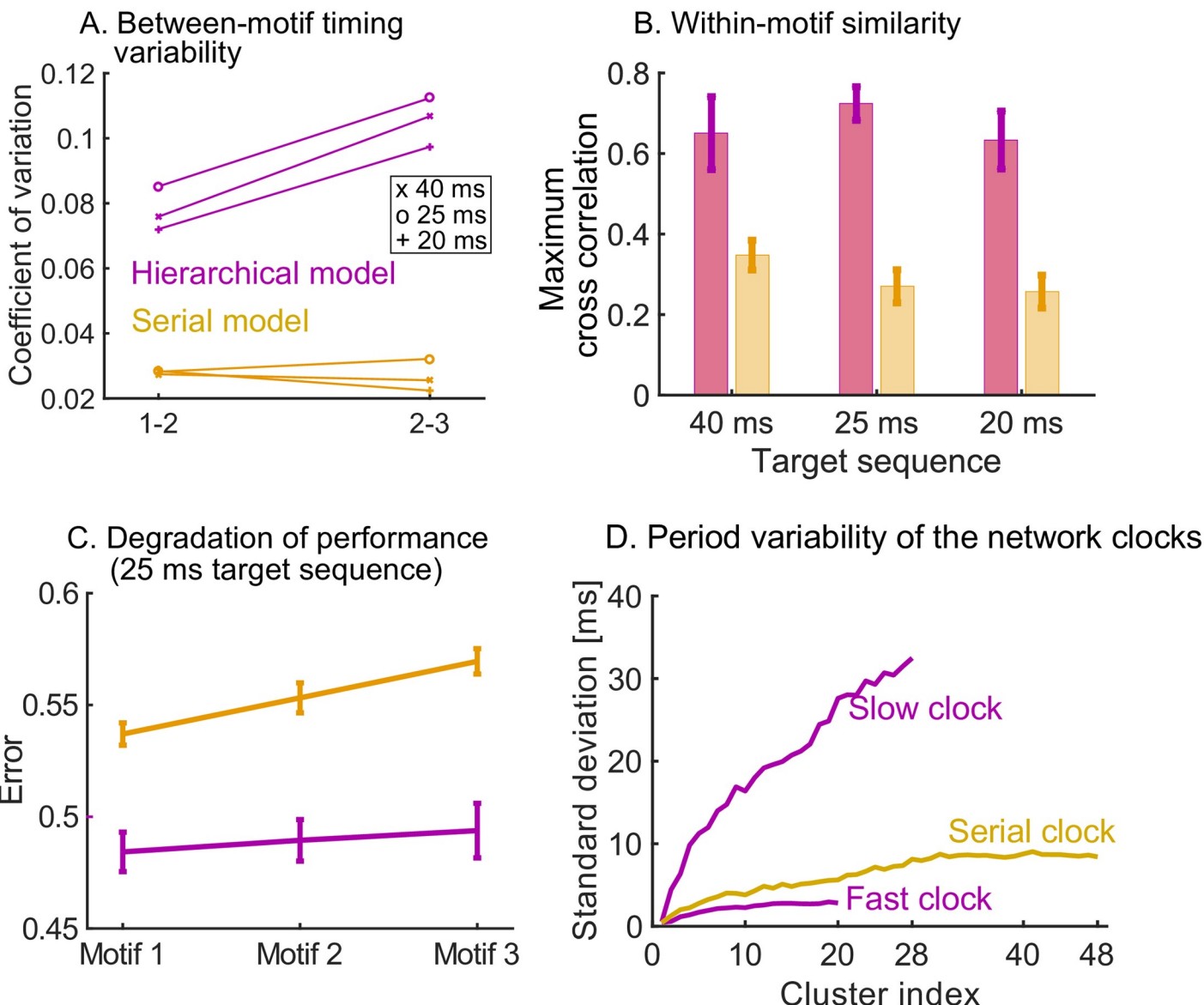

**Fig 5. Measuring variability and performance in the read-out dynamics.** A. The time between motifs 1 and 2 and motifs 2 and 3 is measured during spontaneous dynamics. We plot the coefficient of variation of these times (50 trials) on the y-axis, for the three target sequences in Fig 4A. B. The cross correlation between the spike trains in the first motif and the second and third motif is measured, normalized by the auto-correlation of the spike trains in motif 1. The maximum of the cross correlation is recorded in each trial (50 trials). This is repeated for the three target sequences in Fig 4A. C. We measure the error between the target sequence with 25 ms stimulations in Fig 4A and spike trains in motif 1, 2 and 3. In both models, the performance degrades towards later occurring motifs. The degradation is significantly worse in the serial model: a linear regression yields a slope of 0.0163 for the serial model and a slope of 0.0048 for the hierarchical model ($p < 10^{-5}$ using t-test). D. The serial clock (48 clusters) is obtained by adding the slow (28 clusters) and fast (20 clusters) clocks together. Sequential dynamics is simulated 50 times for each clock. The time at which each cluster is activated in the sequential dynamics is measured. The standard deviation of these activation times is plotted as a function of the cluster index. The serial clock has a maximal variability of about 9 ms. The fast and slow clock have a maximal variability of about 3 and 35 ms respectively.

store multiple sequences to be used as needed. Motivated by this idea, we explore the capacity of the hierarchical model to learn, store and replay more than one sequence, and we compare it to the alternative serial model. We define capacity here as the number of neurons and synapses needed to store a number of sequences $N_S$. First, we note that a new sequence can be stored in the hierarchical model by adding another interneuron network in parallel. The additional

interneuron network is a replica of the existing one, with the same structure and connection weights to the rest of the system.

Each interneuron network learns one syntax, in the same way as one read-out network learns one motif. As an illustration, we learn the sequences *AAB* and *BAAB* (Fig 6A), by

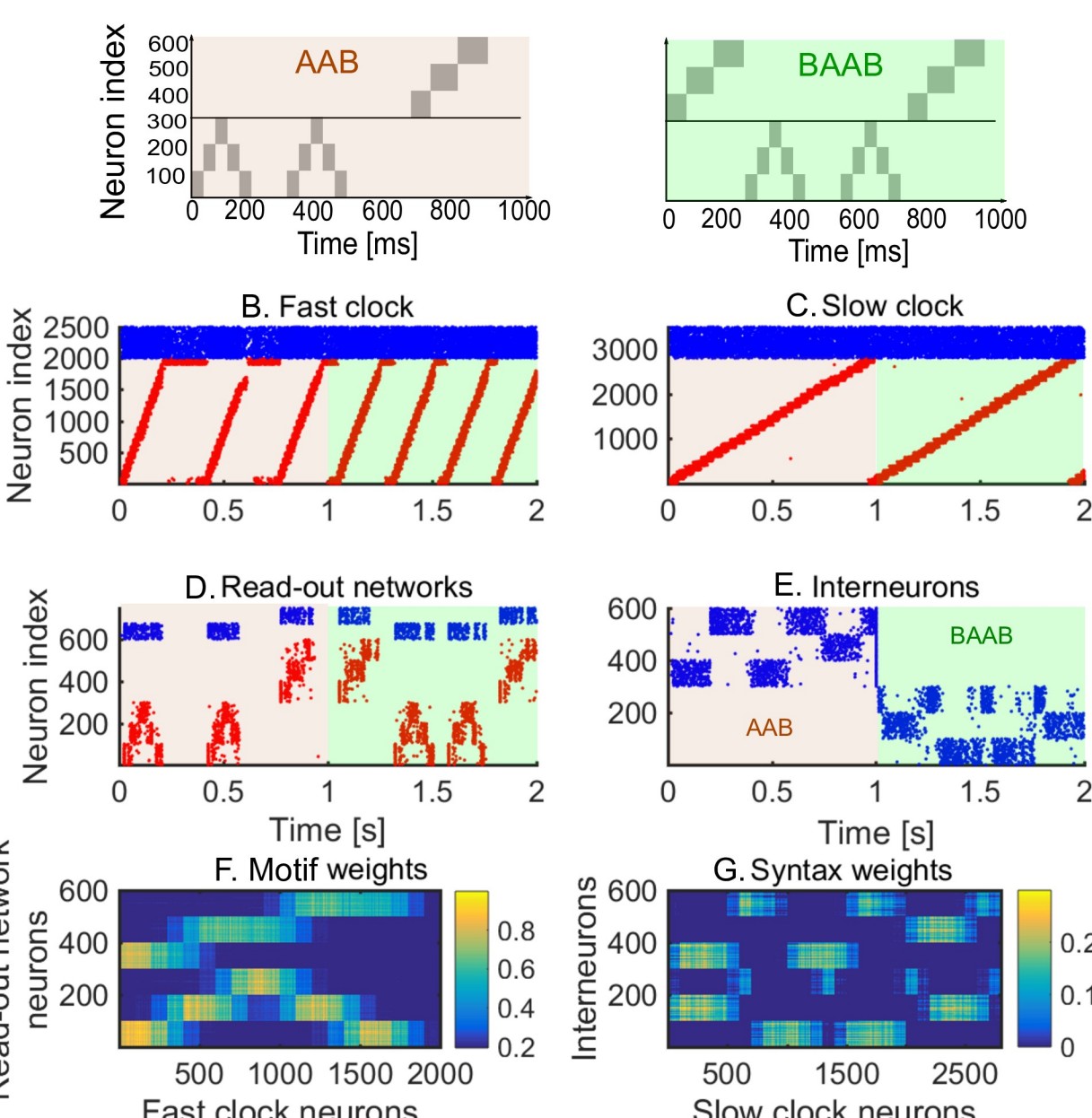

**Fig 6. Spontaneous dynamics after learning two sequences alternately (80 learning iterations).** A. The target sequences. B-E. Red dots: excitatory neurons; blue dots: inhibitory neurons. Brown shaded area: sequence *AAB* is played by inhibiting the interneurons related to the second sequence; light green shaded area: sequence *BAAB* is played by inhibiting the interneurons related to the first sequence. B. Spike raster of the fast clock. C. Spike raster of the slow clock. D. Spike raster of the two read-out networks. E. Spike raster of the interneurons. An external attentional inhibitory current selects which sequence is played. F. The motif weights encode the two motifs. Note the similarity with Fig 2F: the same motifs are re-used in both sequences. G. The syntax weights encode the two motif orderings. Note the difference with Fig 2G: an additional syntax is stored. All motif and syntax synapses are plastic at all times during the sequence presentations.

presenting the target sequences alternately. We then simulate spontaneous dynamics to test that the learning is successful. The spiking dynamics (Fig 6B–6E) show that the model is able to replay the two sequences. To select between the two sequences, we use an attentional external current to the interneuron networks during learning and spontaneous dynamics (shaded areas in Fig 6E). Depending on the interneuron activity, the fast clock (Fig 6B) and read-out networks (Fig 6D) are active. Note that the motifs are encoded in the motif weights (Fig 6F) and syntax weights encode both target motif orderings (Fig 6G). These results show that the hierarchical model can learn, store and replay multiple sequences. Importantly, the motifs are still efficiently re-used: when motifs *A* and *B* are learnt by presenting sequence *AAB*, they can immediately be re-used when a different syntax (e.g., *BAAB*) is presented.

We then compare the efficiency of the hierarchical model to the serial model (S5 Fig). In the serial model, read-out networks have to be added in order to learn and store multiple sequences. This is inefficient for two reasons: 1) The same motif might be stored in different read-out networks, making learning slower; 2) The addition of new read-out networks in the serial model requires more 'hardware' (i.e. neurons and synapses) than the addition of an interneuron network in the hierarchical model. In the case where we have a number of sequences $N_S$ consisting of two motifs (and using network parameters as detailed in the Methods), we have the following capacities. For the serial model, we have 6000 neurons in the serial clock (of which 4800 are excitatory), and 750 neurons in each read-out network (of which 600 are excitatory). Then the number of neurons needed is $6000 + 750 \cdot N_S$, and the number of plastic synapses needed is $4800 \cdot N_S \cdot 600$.

For the hierarchical model, on the other hand, there are 6000 neurons in the fast and slow clocks combined, 750 neurons in the read-out network, and 300 neurons in each interneuron network. Hence the number of neurons needed is $6750 + 300 \cdot N_S$, and the number of plastic synapses and non-plastic lateral connections due to the interneuron network is $2000 \cdot 600 + 2800 \cdot 300 \cdot N_S + 750 \cdot 300 \cdot N_S + 600 \cdot 300 \cdot N_S + 2000 \cdot 100 \cdot N_S + 200 \cdot 100 \cdot N_S = (2000 + 2441.67 \cdot N_S) \cdot 600$. For two sequences ($N_S = 2$), we then have 7500 neurons and 5, 760, 000 synapses for the serial model, whereas the hierarchical model requires 7350 neurons and 4, 130, 000 synapses—a significant reduction in resources. Even when $N_S = 1$, the hierarchical model uses more neurons than the serial model, but still fewer synapses. In general, the hierarchical model scales substantially more favourably as $N_S$ is increased.

Finally, we extend the hierarchical model to learn two sequences consisting of in total six motifs, (S6 Fig). We generalize the model to include multiple motif durations and observe that the hierarchical model is scalable. A serial model would need 10, 500 neurons and 17, 280, 000 synapses. The hierarchical model uses instead 9650 neurons and 13, 630, 000 synapses, again a significant reduction in resources. Our results show that the hierarchical model can learn and store multiple sequences by adding more interneuron networks in parallel. This hierarchical organisation thus exploits the compositional nature of the sequences, in a way the serial model cannot, leading to increased capacity. The hierarchical model primarily uses less synapses.

## The hierarchical model displays increased robustness to perturbations of the sequential dynamics

We next investigate the role of the hierarchy in enhancing the robustness of the model to perturbations in the target sequence. Behavioural perturbation experiments have shown that individual motifs can be removed mid-sequence without affecting later-occurring motifs [13]. This is a useful feature which can dramatically improve the robustness of responses, since later-occurring behaviour does not critically depend on the successful completion of all previous behaviours. To examine this issue, we have carried out simulations on the serial and

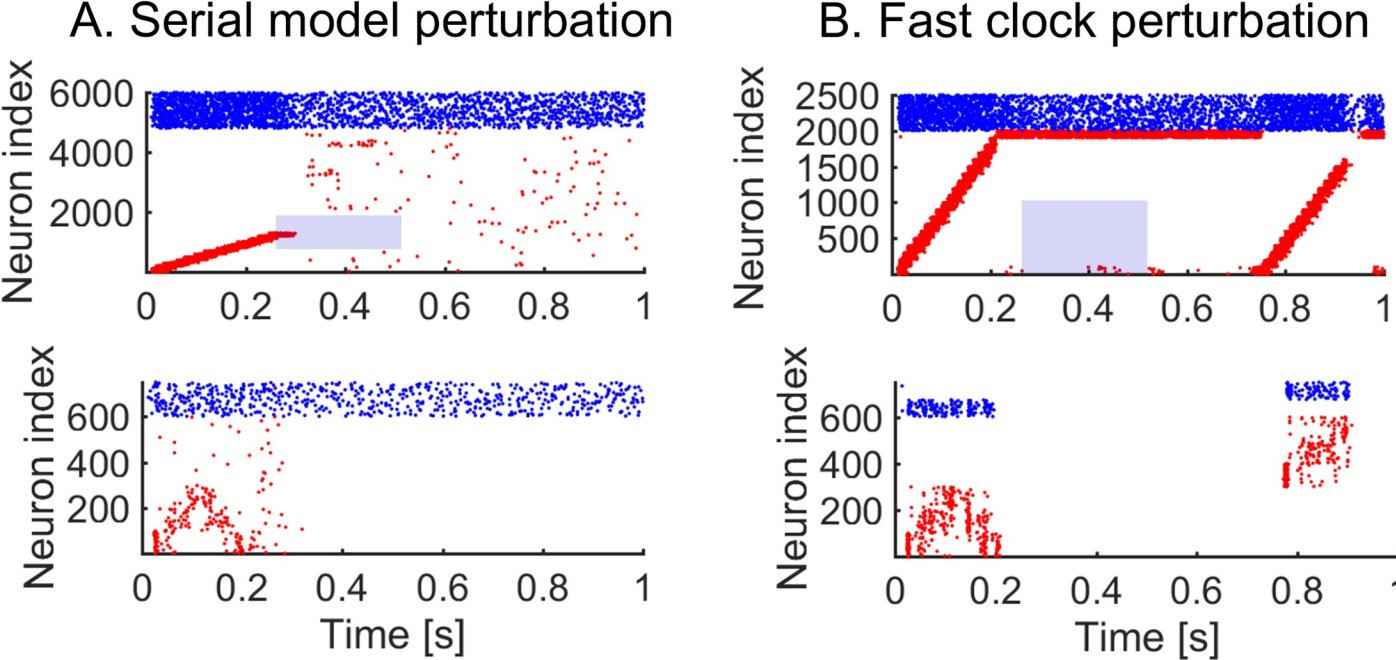

**Fig 7. Perturbing the dynamics.** We learn sequence *AAB* and then apply a perturbation. Blue shade indicates the perturbation time, and neurons perturbed. A. 250 ms perturbation of the serial network clock. The targeted neurons (neurons 1000 to 2000) have no excitatory external input during the perturbation. The sequential activity breaks down completely. B. 250 ms perturbation of the fast clock in the hierarchical model. The targeted neurons (neurons 1 to 1000) have no excitatory external input during perturbation. The sequential activity breaks down but is reactivated for the final motif through the interneurons.

hierarchical models under perturbations in the firing of the neurons in the clock; specifically we remove the external input to excitatory neurons in the clock network. In the serial model, the perturbation leads to the breakdown of the remaining parts of the sequence (Fig 7A), whereas when the same perturbation is applied to the fast clock of the hierarchical model, we see that later motifs are preserved (Fig 7B). The reason is that the dynamics in the slow clock is intact and continues to drive the behaviour. Perturbing the slow clock, and keeping the fast clock intact, has less predictable outcomes for the dynamics. Random activity in the interneurons can cause motifs to be played in a random order (S7 Fig). Overall, the hierarchical model improves the robustness. Indeed, at any point in time, a single cluster of neurons is active in the clock of the serial model, whereas there are two active clusters of neurons (one in the fast clock and another in the slow clock) in the hierarchical model. This separation of time scales is fundamental to preserve the robustness of the model.

## Discussion

### Summary of results

We have presented here a hierarchical neuronal network model for the learning of compositional sequences. We demonstrated how motifs and syntax can be learnt independently of each other. The hierarchical structure has direct implications for the learning and is contrasted with a serial architecture. The hierarchical structure leads to an increased learning speed and the possibility to efficiently relearn the ordering of individual motifs. The replays of individual motifs are more similar to each other as compared to replays in the serial model. Separating the motifs and syntax into two different pathways in the hierarchical model has also implications for the resources used and robustness. The motifs can be re-used

in the hierarchical model, leading to a significant reduction in hardware (i.e. neurons and synapses). Finally, the serial model has a single pathway, as opposed to two, and is therefore more prone to perturbations.

## From serial to hierarchical modelling

Modelling studies so far have either focused on the study of sequential dynamics [38–41] or on motif acquisition [27–29]. This paper introduces an explicitly hierarchical model as a fundamental building block for the learning and replay of sequential dynamics of a compositional nature. Sequential dynamics is omnipresent in the brain and might be important in time-keeping during behaviour [19, 42–46]. When temporal sequences are compositional (i.e., formed by the ordering of motifs), they lead to the presence of different time scales associated with the motifs and their ordering (or syntax). From the perspective of learning, such multiscale temporal organisation lends itself naturally to a hierarchical organisation, where the different scales are associated with different groups of neurons in the network (see also [47]). While sequential dynamics has been observed, coordination between multiple sequences on different scales, as we propose in this paper, has not been observed experimentally. We thus present this as a prediction that sequences on different scales may organize compositional behaviour in the same brain region or across different brain regions.

Hierarchical temporal structures might arise during development in a variety of ways [23, 48]. One way is that a single *proto*sequence is learnt first. The *proto*sequence covers the entire behaviour learning the most crude aspects. This might then be followed by splitting the *proto*sequence into multiple sequences specialized to different aspects of the behaviour. A similar splitting of sequences has been observed in birdsong [49, 50]. Hierarchical motor control has also been studied in the artificial intelligence field [51]. A recent model works towards closing the gap from a machine system to a biological system [52] but remains non-trivial to implement using dynamics and plasticity that are considered to be more realistic in a biological sense.

## Limitations of the hierarchical model

An important aspect of the hierarchical model is the interneuron network, which coordinates the different time scales. The specificity of the hardwired lateral connectivity to and from the interneuron network is a limitation of the model, but does not require extensive fine tuning, as seen in S8 Fig. An important aspect of sequential behaviour is the ability to vary the speed of execution. In the serial model, the speed can easily be controlled by playing the serial clock faster or slower (see also [29, 36]). In the hierarchical model, this is not as straightforward. One possibility is that the fast and slow clock coordinate the increase or decrease in speed. A different possibility could be that the speed is controlled in a network downstream of the read-out network.

## A storage and replay device

The proposed model can be viewed as a biological memory device that stores sequences by means of supervised learning and replays them later by activating the device with spontaneous activity. However, it is important to note that during spontaneous activity there is no input to the device other than the random spike patterns that keep the dynamics of the system going. This mode of operation is therefore distinct from computational machines, such as the liquid state machine [53, 54] or the tempotron [55], where different input patterns are associated with and transformed into different output patterns. Such computational machines, where

spiking patterns are inputs to be transformed or classified, are thus complementary to our autonomous memory device.

## Hierarchies in other tasks

Hierarchies exist beyond the somewhat simple learning of compositional sequences, and it is expected that hierarchical models share common basic features despite solving highly distinct problems. For instance, a recent example of a hierarchical model for working memory uses two different networks: an associative network and a *task-set* network [56]. In our setting, the associative network could be identified with the motifs (fast clock+read-out) whereas the *task-set* network would correspond to the syntax (slow clock+interneurons). Navigation is another typical example of a task where hierarchy is used [57], and the discovery of structure in an environment is closely related to the presence of a hierarchy [58].

## Relating the model to experiments

As mentioned above, there are *qualitative* similarities between the proposed hierarchical model and experimental studies. Experimental studies have pointed increasingly at the importance of hierarchical organisation both in structural studies as well as in the learning and execution of movement and auditory sequences. For example, behavioural re-learning has shown that birds can re-order motifs independently from the within-motif dynamics [22]. Optogenetical perturbation in the striatum of mice has shown that individual motifs can be deleted or inserted mid-sequence, without altering the later part of the behavioural sequence [13]. The proposed model aims to provide a conceptual framework to explain such behavioural observations while simultaneously using biophysically realistic spiking networks and plasticity rules.

However, a *quantitative* link between model and experiment is not trivial. This is true for behaviour, but even more so for neural activity. Indeed, our model has free parameters, including topology and plasticity, which need to be tuned to the task at hand. Nevertheless, there are two recent advances that may help future work in this direction. Firstly, there have been recent technological improvements in recording of behaviour [18, 59] and neural activity [60] along with the possibility to apply perturbations [13]. Secondly, there has been progress in decoding meaningful information from large observational datasets [61], e,g, the extraction of sequences from neural recordings [62] and the analysis of learning behaviour of songbirds [63]. In this vein, an interesting question to pursue is whether one could *rediscover* the hierarchical structure from temporal data generated by our model. For instance, one could observe a randomly chosen subset of neurons in the model: could the hierarchical organisation and function of the network be inferred from those partial observations by using data analysis?

## Conclusion

Using realistic plasticity rules, we built a spiking network model for the learning of compositional temporal sequences of motifs over multiple time scales. We showed that a hierarchical model is more flexible, efficient and robust than the corresponding serial model for the learning of such sequences. The hierarchical model concentrates the variability in the inter-motif timings but achieves high motif fidelity.

## Methods

Excitatory neurons (*E*) are modelled with the adaptive exponential integrate-and-fire model [64]. A classical integrate-and-fire model is used for the inhibitory neurons (*I*). Motifs and

syntax are learnt using simple STDP-rules (see for example [65]) without need for additional fast normalization mechanisms.

## Model architecture

The hierarchical model consists of four recurrent networks. Each network and their parameters are described below. Synaptic weights within each recurrent network are non-zero with probability $p = 0.2$. The synaptic weights in the recurrent networks which produce sequential dynamics are scaled using a scaling factor $f \sim 1/\sqrt{N}$, i.e., it scales with the corresponding network size $N$.

**Fast clock (Fc).**   The fast clock network has $N_{Fc}^E = 2000$ excitatory and $N_{Fc}^I = 500$ inhibitory neurons recurrently connected with parameters shown in Table 1. Sequential dynamics is ensured by dividing the excitatory neurons in 20 clusters of 100 neurons. The baseline excitatory weights $w_{Fc}^{EE}$ within the same cluster are multiplied with a factor of 25, whereas the excitatory weights from cluster $i$ to cluster $i + 1$ mod 20 ($i = 1..20$) are multiplied by a factor of 12.5. Previous studies have shown that such a weight structure leads to sequential dynamics and can be learnt in a biologically plausible way [29, 35, 36]. The last cluster in the fast clock has a special role. It is not inhibited by the 'silent' interneurons and as such remains active during silent periods. Once the silent period is over, it restarts the fast clock by activating the first cluster. The fast clock receives excitatory external random Poisson input and inhibitory input from the interneurons, and projects to the read-out networks.

**Read-out networks (R).**   Each read-out network codes for one individual motif. There are no overlaps or connections between the different read-out networks. The read-out networks are identical and balanced (see Table 2 for the parameters). The excitatory neurons in the read-out networks receive excitatory input from the plastic motif synapses coming from the fast clock. All read-out neurons receive inhibitory input from the interneurons. All read-out neurons also receive external inputs: a supervisor component (only during learning) and a random input. The results are not sensitive to the exact configuration of the read-out networks (see S9 Fig).

**Slow clock (Sc).**   The slow clock network has $N_{Sc}^E = 2800$ excitatory and $N_{Sc}^I = 700$ inhibitory neurons, recurrently connected. It is essentially a scaled copy of the fast clock. Table 3 shows the parameters of this network. Sequential dynamics is ensured by dividing the excitatory neurons in 28 clusters of 100 neurons. The baseline excitatory weights $w_{Fc}^{EE}$ within the same cluster are multiplied with a factor of 25, and the excitatory weights from cluster $i$ to cluster $i + 1$ mod 28 ($i = 1..28$) are multiplied by a factor of 4.7. The slow clock receives excitatory external random Poisson input and projects to the interneuron networks.

**Interneuron networks (In).**   Each interneuron network codes for one syntax. There are no overlaps between the interneuron networks. Each interneuron network is balanced with

**Table 1. Fast clock network parameters.**

| Constant | Value | Description |
|---|---|---|
| $N_{Fc}^E$ | 2000 | Number of recurrent E neurons |
| $N_{Fc}^I$ | 500 | Number of recurrent I neurons |
| $f$ | 0.6325 | Scaling factor |
| $w_{Fc}^{EE}$ | $5f$ pF | Baseline E to E synaptic strength |
| $w_{Fc}^{IE}$ | $3.5f$ pF | E to I synaptic strength |
| $w_{Fc}^{EI}$ | $110f$ pF | I to E synaptic strength |
| $w_{Fc}^{II}$ | $36f$ pF | I to I synaptic strength |

**Table 2. Read-out network parameters.**

| Constant | Value | Description |
|---|---|---|
| $N_R^E$ | 300 | Number of recurrent E neurons |
| $N_R^I$ | 75 | Number of recurrent I neurons |
| $w_R^{EE}$ | 3 p$F$ | E to E synaptic strength |
| $w_R^{IE}$ | 6 p$F$ | E to I synaptic strength |
| $w_R^{EI}$ | 190 p$F$ | I to E synaptic strength |
| $w_R^{II}$ | 60 p$F$ | I to I synaptic strength |

**Table 3. Slow clock network parameters.**

| Constant | Value | Description |
|---|---|---|
| $N_{Sc}^E$ | 2800 | Number of recurrent E neurons |
| $N_{Sc}^I$ | 700 | Number of recurrent I neurons |
| $f$ | 0.5345 | Scaling factor |
| $w_{Sc}^{EE}$ | 5$f$ p$F$ | Baseline E to E synaptic strength |
| $w_{Sc}^{IE}$ | 3.5$f$ p$F$ | E to I synaptic strength |
| $w_{Sc}^{EI}$ | 110$f$ p$F$ | I to E synaptic strength |
| $w_{Sc}^{II}$ | 36$f$ p$F$ | I to I synaptic strength |

parameters given in Table 4. Neurons within each interneuron network are grouped into 3 groups of 100 neurons: one group per motif and one group for the 'silent' motif. The 'silent' motif inhibits all clusters in the fast clock except the last one, and also silences all read-out motifs. The interneuron networks receive excitatory input from all other networks. They also receive random excitatory external input.

**Connections between recurrent networks.** The recurrent networks are connected to each other to form the complete hierarchical architecture. All excitatory neurons from the fast clock project to all excitatory neurons in the read-out networks. These synapses, $w_0^M$, are plastic. All excitatory neurons from the slow clock project to all the interneurons. These synapses, $w_0^S$, are also plastic. To signal the end of a motif, the penultimate cluster of the fast clock activates the interneurons of the 'silent' motif. The last cluster is also connected to the 'silent' motif which silences all other clusters in the fast clock and all neurons in the read-out networks. Each read-out network gives excitatory input to its corresponding interneuron group. This interneuron group laterally inhibits the other read-out network(s). Table 5 gives all the parameters of the connections between the different networks. To understand the limitations of the model, we test a range of lateral connectivity parameters (S8 Fig).

**Serial model (Sm).** The hierarchical model is compared with a serial model. The serial model has one large clock (with the same number of neurons as the fast and slow clocks combined) and no interneurons. Sequential dynamics is generated by clustering the neurons in the network in 48 clusters of 100 neurons. The baseline excitatory weights $w_{Sm}^{EE}$ of the same cluster are multiplied with a factor of 25, and the excitatory weights from group $i$ to group $i + 1$ mod 48

**Table 4. Interneuron network parameters.**

| Constant | Value | Description |
|---|---|---|
| $N_{In}^I$ | 300 | Number of recurrent I neurons |
| $w_{In}^{II}$ | 25 p$F$ | I to I synaptic strength |

**Table 5. Connections between four networks.**

| Constant | Value | Description |
|---|---|---|
| $w_0^M$ | 0.3 p$F$ | Initial motifs synaptic strengths |
| $w_0^S$ | 0.1 p$F$ | Initial syntax synaptic strengths |
| $w^{RIn}$ | 50 p$F$ | In to R synaptic strength of lateral inhibition |
| $w^{RIn}$ | 20 p$F$ | In to R synaptic strength of silencing motif |
| $w^{InR}$ | 0.4 p$F$ | R to In synaptic strength |
| $w^{FcIn}$ | 20 p$F$ | In to Fc synaptic strength of silencing motif |
| $w^{InFc}$ | 1.5 p$F$ | Penultimate Fc cluster to In synaptic strength |
| $w^{InFc}$ | 0.4 p$F$ | Last Fc cluster to In synaptic strength |
| $w^{InFc}$ | 0 p$F$ | Other Fc clusters to In synaptic strength |

($i = 1..48$) are multiplied by a factor of 6. Table 6 shows the network parameters. The read-out network is kept unchanged (Table 2).

## Neural and synaptic dynamics

All neurons in the model are either excitatory ($E$) or inhibitory ($I$). The parameters of the neurons do not change depending on which network they belong to. Parameters are consistent with Ref. [66].

**Membrane potential dynamics.** The membrane potential of the excitatory neurons ($V^E$) has the following dynamics:

$$
\frac{dV^E(t)}{dt} = \frac{1}{\tau^E}\left( E_L^E - V^E(t) + \Delta_T^E \exp\left( \frac{V^E(t) - V_T^E}{\Delta_T^E} \right) \right)
$$
$$
+ g^{EE}\frac{E^E - V^E(t)}{C} + g^{EI}\frac{E^I - V^E(t)}{C} - \frac{a^E}{C}
$$

(1)

where $\tau^E$ is the membrane time constant, $E_L^E$ is the reversal potential, $\Delta_T^E$ is the slope of the exponential, $C$ is the capacitance, $g^{EE}, g^{EI}$ are synaptic input from excitatory and inhibitory neurons respectively and $E^E, E^I$ are the excitatory and inhibitory reversal potentials respectively. When the membrane potential diverges and exceeds 20 mV, the neuron fires a spike and the membrane potential is reset to $V_r$. This reset potential is the same for all neurons in the model. There is an absolute refractory period of $\tau_{abs}$. The parameter $V_T^E$ is adaptive for excitatory neurons and set to $V_T^E + A_T$ after a spike, relaxing back to $V_T$ with time constant $\tau_T$:

$$
\tau_T \frac{dV_T^E}{dt} = V_T - V_T^E.
$$

(2)

**Table 6. Serial model clock network parameters.**

| Constant | Value | Description |
|---|---|---|
| $N_{Sm}^E$ | 4800 | Number of recurrent E neurons |
| $N_{Sm}^I$ | 1200 | Number of recurrent I neurons |
| $f$ | 0.4082 | Scaling factor |
| $w_{Sm}^{EE}$ | 5$f$ p$F$ | Baseline E to E synaptic strength |
| $w_{Sm}^{IE}$ | 3.5$f$ p$F$ | E to I synaptic strength |
| $w_{Sm}^{EI}$ | 110$f$ p$F$ | I to E synaptic strength |
| $w_{Sm}^{II}$ | 36$f$ p$F$ | I to I synaptic strength |

The adaptation current $a^E$ for excitatory neurons follows:

$$\tau_a \frac{da^E}{dt} = -a^E + \alpha(V^E - E_L^E).$$

(3)

where $\tau_a$ is the time constant for the adaptation current. The adaptation current is increased with a constant $\beta$ when the neuron spikes.

The membrane potential of the inhibitory neurons ($V^I$) has the following dynamics:

$$\frac{dV^I(t)}{dt} = \frac{E_L^I - V^I(t)}{\tau^I} + g^{IE} \frac{E^E - V^I(t)}{C} + g^{II} \frac{E^I - V^I(t)}{C}.$$

(4)

where $\tau^I$ is the inhibitory membrane time constant, $E_L^I$ is the inhibitory reversal potential and $E^E$, $E^I$ are the excitatory and inhibitory resting potentials respectively. $g^{EE}$ and $g^{EI}$ are synaptic input from excitatory and inhibitory neurons respectively. Inhibitory neurons spike when the membrane potential crosses the threshold $V_T$, which is non-adaptive. After this, there is an absolute refractory period of $\tau_{abs}$. There is no adaptation current (see Table 7 for the parameters of the membrane dynamics).

**Synaptic dynamics.** The synaptic conductance, $g$, of a neuron $i$ is time dependent, it is a convolution of a kernel with the total input to the neuron $i$:

$$g_i^{XY}(t) = K^Y(t) * \left( W_{ext}^X s_{i,ext}^X + \sum_j W_{ij}^{XY} s_j^Y(t) \right).$$

(5)

where $X$ and $Y$ can be either $E$ or $I$. $K$ is the difference of exponentials kernel:

$$K^Y(t) = \frac{e^{-t/\tau_d^Y} - e^{-t/\tau_r^Y}}{\tau_d^Y - \tau_r^Y},$$

with a decay time $\tau_d$ and a rise time $\tau_r$ dependent only on whether the neuron is excitatory or inhibitory. The conductance is a sum of recurrent input and external input. The externally incoming spike trains $s_{ext}^X$ are generated from a Poisson process with rates $r_{ext}^X$. The externally

**Table 7. Neuronal membrane dynamics parameters.**

| Constant | Value | Description |
| --- | --- | --- |
| $\tau_E$ | 20 ms | E membrane potential time constant |
| $\tau_I$ | 20 ms | I membrane potential time constant |
| $\tau_{abs}$ | 5 ms | Refractory period of E and I neurons |
| $E^E$ | 0 mV | excitatory reversal potential |
| $E^I$ | −75 mV | inhibitory reversal potential |
| $E_L^E$ | −70 mV | excitatory resting potential |
| $E_L^I$ | −62 mV | inhibitory resting potential |
| $V_r$ | −60 mV | Reset potential (both E and I) |
| $C$ | 300 pF | Capacitance |
| $\Delta_T^E$ | 2 mV | Exponential slope |
| $\tau_T$ | 30 ms | Adaptive threshold time constant |
| $V_T$ | −52 mV | Membrane potential threshold |
| $A_T$ | 10 mV | Adaptive threshold increase constant |
| $\tau_a$ | 100 ms | Adaptation current time constant |
| $\alpha$ | 4 nS | Adaptation current factor |
| $\beta$ | 0.805 pA | Adaptation current increase constant |

**Table 8. Synaptic dynamics parameters.**

| Constant | Value | Description |
|---|---|---|
| $\tau_d^E$ | 6 ms | E decay time constant |
| $\tau_r^E$ | 1 ms | E rise time constant |
| $\tau_d^I$ | 2 ms | I rise time constant |
| $\tau_r^I$ | 0.5 ms | I rise time constant |
| $W_{ext}^E$ | 1.6 pF | External input synaptic strength to E neurons |
| $r_{ext}^E$ | 4.5 kHz | Rate of external input to E neurons |
| $W_{ext}^I$ | 1.52 pF | External input synaptic strength to I neurons |
| $r_{ext}^I$ | 2.25 kHz | Rate of external input to I neurons |

generated spike trains enter the network through synapses $W_{ext}^X$ (see Table 8 for the parameters of the synaptic dynamics).

## Plasticity

**Motif plasticity.**  The synaptic weight from excitatory neuron $j$ in the fast clock network to excitatory neuron $i$ in the read-out network is changed according to the following differential equation:

$$\frac{dW_{ij}^M(t)}{dt} = -A_{dep}^M + A_{pot}^M \left( y_i(t)\, s_j(t) \; + \; y_j(t)\, s_i(t) \right). \tag{6}$$

where $A_{pot}^V$ and $A_{dep}^V$ are the amplitude of potentiation and depression, $s_i(t)$ is the spike train of the postsynaptic neurons, and $s_j(t)$ is the spike train of the presynaptic neurons. Both pre- and post-synaptic spike trains are low pass filtered with time constant $\tau_M$ to obtain $y(t)$:

$$\tau_M \frac{dy(t)}{dt} = s(t) - y(t). \tag{7}$$

The synapses from the fast clock to the read-out network have a lower and upper bound $[W_{min}^M, W_{max}^M]$. Table 9 shows parameter values for the motif plasticity rule.

**Syntax plasticity.**  Similar to the motif plasticity rule, the syntax plasticity rule has a symmetric window. The dynamics is as such governed by the same equations, with slightly different parameters:

$$\frac{dW_{ij}^S(t)}{dt} = -A_{dep}^S + A_{pot}^S \left( y_i(t)\, s_j(t) \; + \; y_j(t)\, s_i(t) \right). \tag{8}$$

where $s_i(t)$ is the spike train of the postsynaptic neurons, and $s_j(t)$ is the spike train of the presynaptic neurons. The spike trains are low pass filtered with time constant $\tau_S$ to obtain $y(t)$ (as

**Table 9. Motif plasticity parameters.**

| Constant | Value | Description |
|---|---|---|
| $A_{pot}^M$ | 0.03 pFHz | Amplitude of potentiation |
| $A_{dep}^M$ | $2/3 \times 10^{-6}$ pF | Amplitude of depression |
| $\tau_M$ | 5 ms | Time constant of low pass filter |
| $W_{min}^M$ | 0 pF | Minimum I to I weight |
| $W_{max}^M$ | 1 pF | Maximum I to I weight |

**Table 10. Syntax plasticity parameters.**

| Constant | Value | Description |
|---|---|---|
| $A_{pot}^S$ | 0.025 pFHz | Amplitude of potentiation |
| $A_{dep}^S$ | $0.10 \times 10^{-5}$ pF | Amplitude of depression |
| $\tau_S$ | 20 ms | Time constant of low pass filter |
| $W_{min}^S$ | 0 pF | Minimum I to I weight |
| $W_{max}^S$ | 0.3 pF | Maximum I to I weight |

in Eq 7). The synapses from the slow clock to the interneurons have a lower and upper bound $[W_{min}^S, W_{max}^S]$. Table 10 shows parameter values for the syntax plasticity rule. Note that the time constants are longer than the time constants in the motif plasticity.

## Measuring the error

**Motif error.** The fast clock and motif networks are uncoupled from the slow clock and the interneuron network. We simulate spontaneous dynamics in the fast clock and motif networks by giving an input to the first cluster of the fast clock. We end the simulation after one fast clock sequence is completed. The spike trains of the excitatory neurons in the motif networks are compared to the individual target motifs (e.g. *A* or *B*), which are binary. The spike trains are first convolved using a Gaussian kernel of width $\sim 10$ ms. This gives a proxy to the firing rates of the neurons. The firing rates are then normalized between 0 and 1. Dynamic time warping is finally used to compare the normalized spontaneous dynamics to the target sequence. Dynamic time warping is needed to remove the timing variability in the spontaneous dynamics. We computed dynamic time warping using the built-in Matlab function *dtw*. Dynamic time warping was not used to compute the error in Fig 5.

**The ordering error.** Spontaneous dynamics is simulated using the complete model. The target sequence is now the binary target dynamics of the interneurons. Similarly as described above, the spike trains of the interneurons are convolved and normalized to compute the error with the target using dynamic time warping.

**Total error.** Spontaneous dynamics is simulated using the complete model. The spike trains of the excitatory read-out neurons are compared to a binary target sequence to measure the error during learning. The target sequence is the entire sequence of motifs (e.g. *AAB*). The spontaneous spiking dynamics is convolved and normalized, as described above, to compute the error with the target using dynamic time warping.

## Numerical simulations

**Protocol—Learning.** A start current of 5 kHz is given for 10 ms to the first cluster of the slow clock to initiate a training session. Strong supervising input (50 kHz, see also S8 Fig) to the read-out networks controls the dynamics in the read-out networks. The weights from the read-out networks to the interneurons make sure that also the interneurons follow the target ordering: there is no need for an explicit target current to the interneurons. At the start of each motif the fast clock is activated by giving a strong current of 50 kHz to the first cluster for 40 ms. The high supervisor currents are assumed to originate from a large network of neurons, external to this model.

**Protocol—Spontaneous dynamics.** A start current of 5 kHz is given for 10 ms to the first cluster of the slow clock to initiate a spontaneous replay. The slow clock determines which interneurons are active, together with an external attention mechanism (if multiple sequences

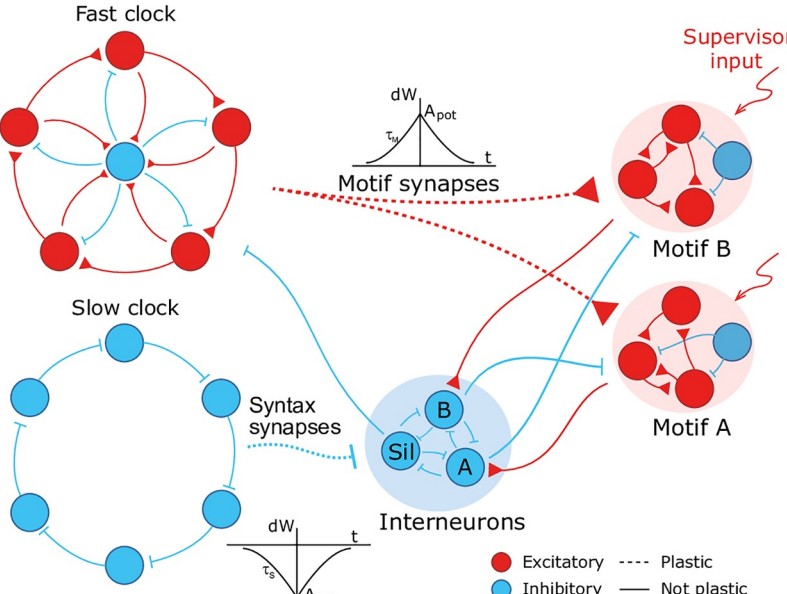

**Fig 8. The networks in the model can have different components.** The slow clock is replaced by an all-inhibitory network (compare with Fig 1). The syntax synapses follow the same STDP rule as the motif synapses, only inverted.

are stored). The interneurons then determine which read-out network is active. The fast dynamics in the read-out networks is controlled by the input from the fast clock.

**Simulations.** The code used for the training and testing of the spiking network model is built in Matlab. Forward Euler discretisation with a time step of 0.1 ms is used. The code is available on GitHub: https://github.com/amaes-neuro/compositional-sequences.

## Changing the slow clock into an all-inhibitory network

The hierarchical model is composed of four networks. These networks can be implemented in various ways. Here, we implement the slow clock differently to illustrate this (Fig 8, to be compared with Fig 1). Sequential dynamics can also be obtained by having an all-inhibitory network (see for example [36]). Learning the sequence *AAB* with this differently implemented hierarchical model leads to similar results (Fig 9, to be compared with Fig 2). Table 11 shows the new slow clock inhibitory network parameters. We conserve the other networks. Sequential dynamics in the slow clock is ensured by grouping the inhibitory neurons in 20 clusters of 100 neurons. The inhibitory weights $w^{II}$ of the same group are multiplied with a factor of 1/30. The inhibitory weights from group $i$ to group $i + 1$ mod 20 ($i = 1..20$) are multiplied by a factor of 1/2. This weight structure does not lead to sequential dynamics by itself, some form of adaptation has to be introduced. To this end, short-term depression is used:

$$\tau_{x_d} \frac{dx_d(t)}{dt} = 1 - x_d(t) \tag{9}$$

where $x_d$ is a depression variable for each neuron in the all-inhibitory network, and $\tau_{x_d} = 200$ ms. This variable is decreased by $0.07x_d(t)$ when the neuron spikes. The outgoing weights of each neuron in the network are multiplied with this depression variable. The slow clock receives excitatory external random Poisson input and projects to the interneuron networks. The syntax synapses follow the same dynamics as Eq 8, but the right hand side of the equation is multiplied by $-1$ (an inverted STDP window). The parameters are summarized in Table 12.

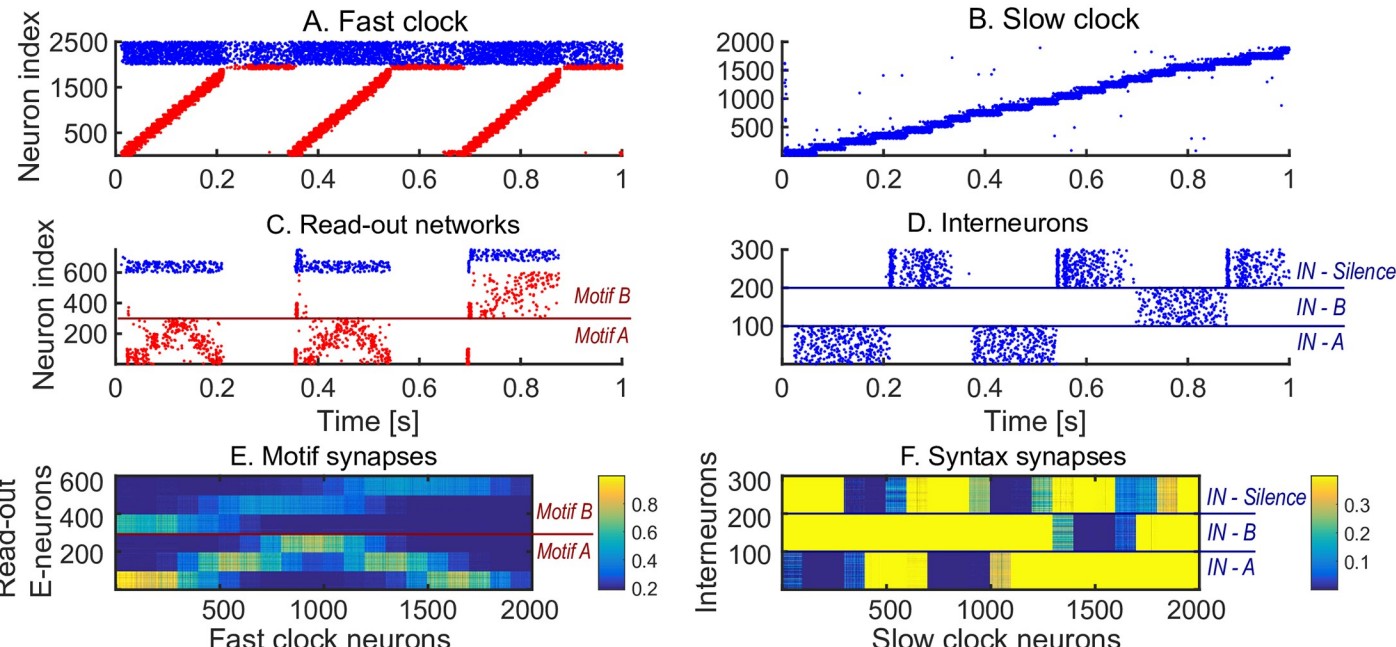

**Fig 9. Learning sequence AAB with an inhibitory slow clock network.** The target sequence is repeatedly presented to the read-out network. A-D. Spontaneous dynamics is simulated after learning (85 target presentations). Red dots: excitatory neurons; blue dots: inhibitory neurons. A. The fast clock, controlled by interneurons 201 to 300. B. The slow clock, consisting of only inhibitory neurons, inhibits the interneurons in the correct order after learning. C. The read-out networks, driven by the fast clock and controlled by the interneurons. D. The interneurons, controlled by the slow clock. E. The motif synapses show that the target motifs *A* and *B* are stored after learning. F. The syntax weights store the correct temporal ordering *A*-silent-*A*-silent-*B*-silent.

**Table 11. Slow clock inhibitory network parameters.**

| Constant | Value | Description |
|---|---|---|
| $N^I$ | 2000 | Number of recurrent I neurons |
| $w^{II}$ | 30 pF | I to I synaptic strength |

**Table 12. Syntax plasticity parameters.**

| Constant | Value | Description |
|---|---|---|
| $A_{pot}^S$ | 0.03 pFHz | Amplitude of potentiation |
| $A_{dep}^S$ | $0.25 \times 10^{-5}$ pFHz | Amplitude of depression |
| $\tau_S$ | 25 ms | Time constant of low pass filter |
| $W_{min}^S$ | 0 pF | Minimum I to I weight |
| $W_{max}^S$ | 0.3 pF | Maximum I to I weight |

## Supporting information

**S1 Text. Extending the model to more and variable motif lengths.** We extend the model such that it can learn sequences consisting of more than two motifs, with variable durations. In the main text, each motif has the same duration. This means the supervisor only needs to provide a starting signal to the fast clock, indicating when a motif starts. In general, a motif can be shorter than a fast clock sequence. In that case, the supervisor has to provide a stop signal to

the fast clock, indicating when a motif ends. This stop signal activates the penultimate cluster in the fast clock, which activates in turn the 'silent' interneurons. The stop signal is 10 ms long and has the same rate as the start signal. We learn example sequences to illustrate this (S6 Fig). Specifically, we learn sequences *ABCD* and *EBCF*. Motifs *A* and *B* are both 200 ms long. Motif *C*, *D*, *E* and *F* are respectively 150 ms, 120 ms, 180 ms and 100 ms long. To keep the sequences as general as possible, we also include variable inter-motif intervals. The silent gap between motifs *A* and *B*, motifs *B* and *C*, and motifs *C* and *D* is respectively 70 ms, 50 ms, and 80 ms. The silent gaps in the second sequence between motifs *E* and *B*, motifs *B* and *C*, and motifs *C* and *F* are respectively 70 ms, 50 ms, and 150 ms. We observe that the model is able to learn the two sequences, but the replay of shorter motifs *D* and *F* is less accurate. The parameters used in this simulation are the same as in other simulations, with an increased network size for the interneuron networks, and read-out network.
(PDF)

**S1 Fig. Dynamics of the hierarchical model during target sequence presentation.** A. The first cluster of the fast clock receives a high input current at the start of each motif presentation. B. The first cluster of the slow clock receives a high input current at the beginning of the sequence presentation. C. The high input current forces spiking in the read-out neurons. D. The read-out neurons activate the interneurons.
(TIF)

**S2 Fig. The serial network model.** A. A single recurrent network clock (left) produces sequential dynamics and drives the dynamics in the read-out networks (right). The weights from the serial clock to the read-out network are plastic. B. We learn target sequence *AAB*. Spontaneous dynamics is simulated after 90 target sequence presentations. C. The read-out weights after learning. Both motif and syntax information are stored in the same weights.
(TIF)

**S3 Fig. Total sequence error for hierarchical and serial model, during relearning: *AAB* → *ABA*.** Spontaneous dynamics is simulated every fifth training iteration and compared with target sequence *AAB* (brown line) and target sequence *ABA* (dark green line) to compute the total sequence error. A. Total sequence error for the hierarchical model. Note how the total sequence error (which is the combination of within-motif error and syntax error) relative to *AAB* decreases for about 30 iterations after target *ABA* is presented for the first time due to the continued improvement in the within-motif dynamics. After this, there is a marked increase in the syntax error and the total error relative to *AAB*. B. Total sequence error for the serial model. The lack of hierarchy in the serial model implies that both the within-motif dynamics and motif ordering has to be relearned. This leads to a more gradual and slower relearning (note the longer x-axis).
(TIF)

**S4 Fig. Total sequence error for various learning rates, during relearning *AAB* → *ABA*.** Spontaneous dynamics is simulated every fifth training iteration and compared with target sequence *AAB* (brown line) and target sequence *ABA* (dark green line) to compute the total sequence error. The lines shows the average of 5 simulations. A. The solid line shows the same total error as in S3(A) Fig (the baseline). The dashed line shows the total error, when learning faster. The right hand side of Eq 8 is multiplied by a factor 2. B. The total error when learning slower. The right hand side of Eq 8 is divided by a factor 2. More iterations are shown because the model needs more time to learn the sequences.
(TIF)

**S5 Fig. Learning two sequences.** The hierarchical model requires an additional interneuron network. An external current is assumed to inhibit the interneurons for sequence *BAAB* when sequence *AAB* is presented and vice versa. The serial model duplicates the entire read-out network. Here also, an external current is assumed to inhibit the read-out networks for sequence *BAAB* when sequence *AAB* is presented and vice versa.
(TIF)

**S6 Fig. Learning 2 sequences with variable motif durations and variable inter-motif intervals.** A. The two target sequences. Individual motifs have durations between 100 and 200 ms. Inter-motif intervals range from 50 to 150 ms. B-E. Red dots: excitatory neurons; blue dots: inhibitory neurons. Brown shaded area: sequence *ABCD* is played by inhibiting the interneurons related to the second sequence; light green shaded area: sequence *EBCF* is played by inhibiting the interneurons related to the first sequence. B. Spike raster of the fast clock. C. Spike raster of the slow clock. D. Spike raster of the six read-out networks. E. Spike raster of the interneurons. An external attentional inhibitory current selects which sequence is played. F. The motif weights encode the six motifs. Note that motifs *B* and *C* are learned more as they occur in both sequences. G. The syntax weights encode the two sequences. All motif and syntax synapses are plastic at all times during the sequence presentations (see S1 Text for Method details).
(TIF)

**S7 Fig. Perturbing the slow clock of the hierarchical network.** Blue shade indicates the perturbation time, all excitatory neurons receive no external input for 250 ms. The sequential dynamics in the slow clock breaks down (top right) but random activity in the interneurons (bottom right) leads to sequences in the fast clock (top left), which in turn leads to motif replays (bottom left).
(TIF)

**S8 Fig. Limitations on parameters.** A. Spontaneous dynamics is simulated for a range of parameters, for a model that has learned sequence *ABA*. The potentiated motif synapses have values between 0.7 pF and 1 pF. Raster plots of the read-out network is shown. The lateral inhibition $w^{RIn}$ and the lateral excitation $w^{InR}$ are varied. When the lateral inhibition is too weak, the motifs occur at the same time (top left panel). When the lateral inhibition is sufficiently strong, the motifs are replayed well (bottom right panel). B. A supervisor gives input *ABA* to the read-out network, for a model that has stored sequence *AAB*. When the supervisor input is too low (left panel), the stored sequence dominates the dynamics in the read-out network and there will be no relearning. When the supervisor input is sufficiently high (right panel), the stored sequence is overwritten by the supervisor input and there will be relearning.
(TIF)

**S9 Fig. Learning curves for different read-out configurations.** The read-out network in the main text consists of two separate networks, which are not interconnected. A. Cartoon of read-out network without recurrent excitatory connections. B-D: The learning curves when the recurrent connections in the two separate motif networks are zero. The same relearning protocol as in Fig 3 and S4 Fig is used. E. Cartoon of read-out network when the two motif networks are combined and interconnected into one network. F-H: The learning curves when the two motif networks are combined and interconnected into one network. In this case, the same connections as listed in Table 2 are used but multiplied by $1/\sqrt{2}$, and $N_R^E = 600, N_R^I = 150$. The sparsity of the connections remains $p = 0.2$. The same relearning protocol as in Fig 3 and S4 Fig is used.
(TIF)

## Acknowledgments

We thank Victor Pedrosa and Barbara Feulner for helpful comments.

## Author Contributions

**Conceptualization:** Amadeus Maes, Mauricio Barahona, Claudia Clopath.

**Formal analysis:** Amadeus Maes.

**Investigation:** Amadeus Maes, Mauricio Barahona, Claudia Clopath.

**Methodology:** Amadeus Maes, Mauricio Barahona, Claudia Clopath.

**Software:** Amadeus Maes.

**Supervision:** Mauricio Barahona, Claudia Clopath.

**Visualization:** Amadeus Maes, Mauricio Barahona, Claudia Clopath.

**Writing – original draft:** Amadeus Maes, Mauricio Barahona, Claudia Clopath.

**Writing – review & editing:** Amadeus Maes, Mauricio Barahona, Claudia Clopath.

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
