## [Decision Letter · Decision Letter 0]

8 Dec 2020

Dear Dr. Clopath,

Thank you very much for submitting your manuscript "Learning compositional sequences with multiple time scales through a hierarchical network of spiking neurons" for consideration at PLOS Computational Biology.

As with all papers reviewed by the journal, your manuscript was reviewed by members of the editorial board and by several independent reviewers. In light of the reviews (below this email), we would like to invite the resubmission of a significantly-revised version that takes into account the reviewers' comments.

In particular, the revision should be explicit about critical model assumptions and should clarify both the predictions on network structure and the compatibility with current experimental findings.

We cannot make any decision about publication until we have seen the revised manuscript and your response to the reviewers' comments. Your revised manuscript is also likely to be sent to reviewers for further evaluation.

Sincerely,

Abigail Morrison

Associate Editor

PLOS Computational Biology

Samuel Gershman

Deputy Editor

PLOS Computational Biology

Reviewer's Responses to Questions

**Comments to the Authors:**

Reviewer #1: This paper presents a computational model of sequence generation. The novel aspect of the model is that it focuses on hierarchical sequences—such as the sequences of syllables within a motif, and the sequence of motifs (syntax) that compose a songbirds song. The model assumes two clocks, a fast and slow clock responsible for the syllable sequence within a motif and the motif sequence, respectively. There is a readout network for each motif and an interneuron network for each overall sequence (syntax). Learning is supervised but biological, and takes place at the fast clock -> to read-out synapses and slow clock -> interneurons synapses. The problem being addressed is important and the model captures a number of interesting biological properties in a flexible manner. But the model is also require some significant assumptions and is fairly hardwired in some regards, for example, in regard to the connectivity between from the read-out to interneuron networks.

To allow the reader to better understand the limitations of the model and what has to be improved upon in future models the authors should discuss or address the shortcomings of the model and where it deviates from experimental data, including:

1. It appears that, as presented each motif (or the inter-motif interval) has to be the same duration. This would appear to imply a periodicity to the motifs which does not seem to be anecdotally consistent with many sequential behaviors, or with the birdsong data in which different motifs can have different durations.

2. I’m not aware of any evidence that support the notion of two separate and independent fast and slow clocks in the biological literature. For example, I don’t think there is any evidence of a fast and slow sequence/clock in the birdsong system. Do the authors consider this to be a prediction of their model?

3. The specific hardwiring of the readout to interneuron connections along with the presence of essentially independent motif network seems like a potential challenge to the biological plausibility.

Clarify whether the Inh units in the read-out networks receive external input, and plastic synapses from the fast clock.

The model will be easier to understand if in Figure 1 the projections from Motif A and Motif B, and Slow clock are shown to project exclusively and specifically to the appropriate population of units in the interneuron network. Otherwise it is difficult to understand how the supervised learning governs plasticity in the interneuron network.

In Figure 2 the authors should also show the weight matrices of the nonplastic weights from the read-out to interneuron networks, and vice-versa, as well as the interneuron to fast clock. This will greatly help the reader understand how the model works.

Clarify what the multiple dots in each iteration in Figure 3 represent.

Reviewer #2: In their manuscript the authors address the interesting problem of storing sequences (e.g. ABA) that are made up two motifs (A and B), where each motif is again a sequence (e.g. 12321 or 456). A sequence (called syntax) unfolds on a timescale of a second and each motif lasts up to a couple hundred milliseconds. Building on their recent work (Maes, Barahona, Clopath, 2020, Plos Comp. Biol.), Maes et al introduce a spiking neural network structure consisting of four interconnected components: a slow clock, a fast clock, interneuron network, and read out networks. With this network structure they learn motifs in feedforward weights from the fast clock to read out networks, with each read out network representing one motif. Feedforward weights from the slow clock to the interneuron network switch between predefined interneuron clusters. These clusters control which read out network may become active, suppressing all others. They show that this network structure learns the motifs and syntax synapses independently. Further, compared to a single network (serial model) their network structure is more resistant to perturbations. New combinations of the same motifs can be achieved without adding more readout neurons; however, this requires recruiting copies of the inhibitory network. They also find that the pattern of variability differs when compared to a single network as there is less within motif variability however more variability in the timing of motif switching, due to variability in the period of the slow clock.

Overall this is a nice extension of their recent paper (Maes et al, 2020, Plos Comp. Biol.) to now include sequences of sequences on multiple timescales. Previously they used one clock network to control read outs. Switching between read out networks was done manually. The novel aspect here is that a second slower clock is added to control interneuron groups which can switch between read out networks, forming a slow sequence (syntax) of faster subsequences (motif). There remain however concerns about the clarity of presentation, robustness of the architecture, and some of the conclusions drawn, as listed in the following.

Major comments:

- It seems that the model requires specific connectivity and balance between the different weights to achieve the desired behavior. For example, there are specific feedback connections from each motif to one interneuron cluster. Activation of this interneuron cluster recruits lateral inhibition onto all competing motifs. This inhibition must be strong enough to counteract all feedforward excitation from the fast clock to the read out, in the manuscript these inhibitory weights are two orders of magnitude stronger (0.3 pF vs. 50 pF, Table 5). Thus, at least some discussion of the constraints in connectivity, weights, and activity should be added (e.g. ratio of inh/exc weights allowed? How sparse can activity in the motif be before feedback to its interneuron cluster does not recruit sufficient lateral inhibition?). In addition, limitations as well as experimental predictions stemming from these constraints might be helpful to discuss.

- While the idea is compelling, the results section on capacity of the model, the shown results do not sufficiently support the claim. Given that you are talking about how many neurons and synapses need to be added to store a new sequence, it may be more appropriate to discuss and quantify the resources required instead of capacity.

- If understood correctly, if A is repeated at different positions in a sequence (e.g. ABA), or is in different sequences (e.g. AAB vs. BAAB), different representations are required for each instance of A. While the motif for A is preserved, different instances of A are stored in the corresponding weights from slow clock clusters to the inhibitory networks. If this is correct, slow clock clusters can be reused for each instance of A, however different sets of (slow clock to inhibitory network) synapses are required to store multiple representations of A. This strategy seems to save on neurons at the expense of requiring more synapses. Please discuss.

- I agree with the authors when they write ``Non-Markovian sequences are generally hard to learn, because they require a memory about past dynamics". As I understand it, in this work, for a repeating motif in a sequence (e.g. ABA), two different clusters in the slow clock potentiate their projections with interneuron cluster A. The slow clock is used as an index. Perhaps this is a matter of interpretation, but it seems that this bypasses the need for memory about past dynamics. Instead the order of the sequence elements is stored via the slow clock cluster order and is executed through feedforward weights to the interneuron groups, making the problem Markovian. I would be interested to hear the authors' interpretation of this.

- Related to the previous comment: can your system learn and recall overlapping sequences such as ABCD vs. EBCF?

Minor comments:

- Related to point 1, Figure 1 could better reflect the specific connectivity required. For instance, it would be helpful to label the interneurons in the interneuron network. In this case, one belongs to motif A, one to motif B, and one to the fast clock. It would also be helpful to show that the connectivity is predetermined in a very specific way.

- For motif error, is each motif played out independently of the slow clock sequence order? What is exactly stimulated and how?

- Regarding relearning syntax sequences while reusing motifs: which constraints are there on the timescales? Is it correct that the syntax must be learned faster to avoid motifs being relearned?

- Based on Figure 3B, it looks like learning is faster than relearning. Is it possible to speed up the relearning? Does this depend on the timescale of the depotentiation term?

- For movement control, the variation of speed is an essential part. Please discuss, if and how you could control the recall speed in the proposed system.

Reviewer #3: The manuscript by A. Maes et al. presents an interesting spiking neural model for the learning of dynamical spike sequences that are modular, i.e. composed of subsequences. The basic idea is that two clock-like networks, operating at slow and fast speed, govern the progress of the overall sequence and of all individual subsequences, respectively. The selection of a subsequence happens by inhibition of the other subsequences. The responsible interneurons receive input from the slow clock; their input connections are trained such that the right networks are inhibited at the right times. The connections from the fast clock to the different output neuron networks are similarly trained to generate their individual subsequences.

Remarkably, the networks can flexibly relearn the order of subsequences while the precision of individual ones still increases (Fig. 3). This is in contrast to a purely serial model with a single clock. Further, the networks can easily learn sequences where subsequences repeat. The dynamics are robust against deletion of individual subsequences and the slow clock can be implemented with a sequence generating mechanism that is generically slow as it driven by short term depression.

The manuscript is interesting, clear and well-written. I recommend its publication essentially as it is and would like to ask only a few minor questions:

1. In Fig. 1, might it increase clarity to explicitly display the inhibitory connections from IN-A to the network Motif B and from IN-B to the network Motif A?

2. Is the recurrent excitation in the Motif networks important?

3. What resets the fast clock to start with its first group of neurons after inhibition from IN-Silence stops? Fig. 2B seems to indicate that while IN-Silence is active, the last group of the fast network stays active and initiates the first group as soon as inhibition stops.

**Have all data underlying the figures and results presented in the manuscript been provided?**

Reviewer #1: Yes

Reviewer #2: Yes

Reviewer #3: Yes

PLOS authors have the option to publish the peer review history of their article (what does this mean?). If published, this will include your full peer review and any attached files.

Reviewer #1: No

Reviewer #2: **Yes: **Christian Tetzlaff

Reviewer #3: No
---

## [Decision Letter · Decision Letter 1]

8 Mar 2021

Dear Dr. Clopath,

We are pleased to inform you that your manuscript 'Learning compositional sequences with multiple time scales through a hierarchical network of spiking neurons' has been provisionally accepted for publication in PLOS Computational Biology.

Best regards,

Abigail Morrison

Associate Editor

PLOS Computational Biology

Samuel Gershman

Deputy Editor

PLOS Computational Biology

Reviewer's Responses to Questions

**Comments to the Authors:**

Reviewer #1: The authors have done a good job addressing my concerns, and I think the paper is appropriate for publication.

Reviewer #2: The authors have clarified all my points. Thank you.

Reviewer #3: I thank the authors for their careful answering of my questions. I can fully recommend acceptance of the paper.

**Have all data underlying the figures and results presented in the manuscript been provided?**

Reviewer #1: Yes

Reviewer #2: None

Reviewer #3: Yes

PLOS authors have the option to publish the peer review history of their article (what does this mean?). If published, this will include your full peer review and any attached files.

Reviewer #1: No

Reviewer #2: **Yes: **Dr. Christian Tetzlaff

Reviewer #3: No

---

## [Editor Report · Acceptance letter]

19 Mar 2021

PCOMPBIOL-D-20-01730R1 

Learning compositional sequences with multiple time scales through a hierarchical network of spiking neurons

Dear Dr Clopath,

I am pleased to inform you that your manuscript has been formally accepted for publication in PLOS Computational Biology. Your manuscript is now with our production department and you will be notified of the publication date in due course.

With kind regards,

Alice Ellingham
